# Dye-Sensitized Solar Cell (DSSC): Effects on Light Quality, Microclimate, and Growth of *Orthosiphon stamineus* in Tropical Climatic Condition

N. Roslan [1] , M. E. Ya'acob [1,2,*] , D. Jamaludin [3], Y. Hashimoto [4], M. H. Othman [1] , A. Noor Iskandar [1], M. R. Ariffin [5] , M. H. Ibrahim [6], J. Mailan [1], A. H. Jamaluddin [7] , M. F. Mail [8], B. S. N. Aliah [8] and L. Lu [9]

1   Department of Process & Food Engineering, Faculty of Engineering, Universiti Putra Malaysia,
    Serdang 43400, Malaysia; nadirahroslan1010@gmail.com (N.R.); hafiz.othman@gmail.com (M.H.O.);
    ar525rahman@gmail.com (A.N.I.); junaidahmailan1996@gmail.com (J.M.)
2   Centre for Advance Lightning, Power and Energy Research (CAPER), Faculty of Engineering,
    Universiti Putra Malaysia, Serdang 43400, Malaysia
3   Department of Biological & Agriculture Engineering, Faculty of Engineering, Universiti Putra Malaysia,
    Serdang 43400, Malaysia; diyana_upm@upm.edu.my
4   Research Initiative for Supra Materials, Shinshu University, Wakasato 4-17-1, Nagano 380-8553, Japan;
    hashimt@shinshu-u.ac.jp
5   Institute of Advanced Technology, Faculty of Engineering, Universiti Putra Malaysia,
    Serdang 43400, Malaysia; mohdruzaimi@gmail.com
6   Department of Biology, Faculty of Science, Universiti Putra Malaysia, Serdang 43400, Malaysia;
    mhafiz_ibrahim@upm.edu.my
7   School of Mathematics and Statistics, University of New South Wales, Sydney, NSW 2052, Australia;
    ahmadhakiimjamaluddin@gmail.com
8   Engineering Research Centre, Malaysian Agricultural Research and Development Institute (MARDI),
    Serdang 43400, Malaysia; fazlym@mardi.gov.my (M.F.M.); aliah@mardi.gov.my (B.S.N.A.)
9   Department Electrical and Electronics Engineering, Faculty of Engineering, Universiti Putra Malaysia,
    Serdang 43400, Malaysia; engine2018li@gmail.com
*   Correspondence: fendyupm@gmail.com

**Abstract:** The main challenge facing greenhouse designers is to achieve environment-appropriate greenhouses, especially in tropical regions. The excess radiant energy transmitted into the greenhouse predisposes plants to photo-inhibition and consequently reduces crop production. Lately, photovoltaic (PV) modules are equipped as a greenhouse rooftop to minimize the level of irradiation and air temperature in the greenhouse, simultaneously improving its energy consumption. Nevertheless, due to the low level of irradiation, denser conventional PV internal shading would influence the cultivated crops' growth. Thus, Dye Sensitized Solar Cell (DSSC) possesses several attractive features such as transparent, sensitive to low light levels, and various color options that render DSSC a perfect choice able to serve substantially in energy buildings. This study assessed the microclimate conditions inside the greenhouse with semi-transparent DSSC mounted on top of it, describing the Photosynthetic Photon Flux Density (PPFD) ($\mu$mol m$^{-2}$ s$^{-1}$), Vapor Pressure Deficit VPD (kPa), relative humidity (%), and also temperature (°C). The Overall Thermal Transfer Value (OTTV), which indicates the average thermal energy transmission rate across the external layer of a structure envelope, is also presented. The effects of colored DSSC in altering the spectral of sunlight in reference to the *Orthosiphon stamineus* growth responses were determined. The information of the condition of DSSC greenhouse microclimate helps to identify the information for designing PV greenhouses and to produce income from both electric power and agronomic activity.

**Keywords:** PV greenhouse; DSSC; microclimate; *Orthosiphon stamineus*; tropical climate; semi-transparent PV

## 1. Introduction

The structured and systematized agricultural environment has become the preferred method to alleviate mother nature's direct effect, such as the weather and climate change. Such systems (e.g., greenhouses) create the optimal microclimate condition in order to acquire greater crop yields while retaining minimum energy and overhead costs [1]. The microclimate of the greenhouse is described by a set of climatic attributes, which in a way are different from the natural weather conditions, for example, relative humidity, temperature, solar radiation, and carbon dioxide ($CO_2$) concentrations [2]. However, these attributes or parameters exist in the greenhouse and the natural settings are strongly related to one another.

The biggest problem in agriculture today is the discovery of energy supplies that are safe and renewable. The closest solution is photovoltaic (PV) technology, or also known as electrical energy generation via solar that costs next to nothing, and has been one of the green and abundant resources particularly for tropical countries like Malaysia [3]. Located in the equatorial region, Malaysia is a tropical climate country, with daily sunshine hours up to 8.7 h a day and throughout the year [4]. Nevertheless, during summer months, extreme temperatures imposed by direct solar radiation causes adverse effect on agricultural production [5]. Thus, the application of PV modules embedded in agricultural environments or greenhouses as the rooftop is an ingenious and energy-saving method to overcome the severe solar radiation as well as control the greenhouse's air temperature to be at least near the optimum microenvironment needs of plant. In regard to the concept of agro-technology, Othman et al. [6] studied the cultivation of high-value herbal crop, *Orthosiphon stamineus* or also known as Misai Kucing (love shading plant), under the unoccupied PV arrays of a solar farm, which gives a huge profit return. Apart from that, this practice of plotting herbals beneath the PV structures can act as a cooling mechanism with significant carbon reduction outcomes. Nevertheless, shading induced by stationary conventional PV modules (e.g., crystalline silicon) notably harms farm production and the greenhouse microclimate since the crystalline silicon PV module is opaque to sunlight. This type of shading demonstrates the relationship between PV roofs and plants to be an adversary. Sunshade distribution of PV panels above the greenhouse is linearly related to the coverage ratio. To understand this matter, Cossu et al. [7] investigated the greenhouse microclimate that was 50% covered with PV roofs. The results indicated that the availability of PV greenhouse sunlight fell 64% relative to PV free condition, while it was hotter than the outside temperature on the average of 2.8 °C. It was reported that relative humidity was decreased when the temperature increased inside the PV greenhouse. In regards to those technical predicaments, there were extensive studies conducted on fixing PV panels on top of greenhouse roof [8,9], PV greenhouse orientation [10–12], flexible PV panels [13], covering percentage by PV panels [8,14], taller design of PV structures, the suitable crops cultivated under PV modules [15], and the application of translucent PV technologies [16–19] to improve microclimate condition for the benefit of both agricultural and electricity production.

Semi-transparent Dye-Sensitized Solar Cell (DSSC), the third series of innovation for solar PV technology, is an ideal choice in PV greenhouse due to its variation in color and transparency [20], low fabrication cost [21], flexibility in scaling [21], low light level sensitivity [22], and configured for large scale applications [23]. Roslan et al. [24] presented a new greenhouse integrated with DSSC of various colors (altered by dye color) acting as a photoselective shading to alter the greenhouse light spectrum. In this set up, plant growth can be optimized as it allows photomorphogenesis and photosynthesis to occur. In Greece, Ntinas et al. [19] investigated the performance and quality of tomato cultivated in a DSSC greenhouse. The medium-sized tomato inside the greenhouse showed weak productivity, chlorophyll content, and photosynthetic rate compared to the conventional greenhouse. However, the cherry and medium-sized tomato demonstrated a substantially higher bioactive compound of 6% to 26%, specifically their ascorbic acid, lycopene, β carotene, and total carotenoids concentration. In Taiwan, Kuo et al. [25] evaluated the ideal

spectrum and illuminance distributions of different types of PV modules against plant growth, such as luminous panels, multicolored panels, and transparent panels. The findings showed that the broadest spectrum of translucent light was in the range of 550–600 nm for luminous panels, 600–700 nm (multicolored panels), and 480–600 nm (transparent panels), respectively. Therefore, the author concluded that the types of PV modules should be selected appropriately to harness sunlight optimally for plant growth (PAR wavelength: 400–700 nm), whereas other wavelengths to be utilized in generating electricity.

Light is an energy source and a major regulator of plant life. Light (quantity, quality, direction, and periodicity), and other environmental metrics help plants respond to environmental state [26]. For that reason, plants can possibly stimulate physiological, biochemical, and morphological changes going to sustain their presence in the current environmental conditions. The phytochrome, cryptochrome, and phototropin catch signals emitted from lights—blue, red, and far-red light spectrum regions—are essential for this phase [27,28]. The photoselective shading that alters the spectral of sunlight was extensively studied on horticultural crops [29–31], whereby, it can be achieved by using colored shade netting, colored fluid-roof system, and photoselective films (incorporated with pigments or dies). The photoselective shading influence on crops was studied using different shade nets color. Dissimilar to the typical black shade nets, the red and yellow shades significantly stimuli vegetative development, while dwarfing under the blue shade. Alternatively, the grey shade net (absorbing radiation from infrared and near-infrared) improves branching and makes plants bushy, specifically for smaller leaves and fewer variety plants [32]. A previous study has also shown that red light effectively improves photosynthesis and broadens leaf area for common grape vine (*Vitis vinifera*) and leaves biomass compared with sunlight [33].

Meanwhile, in Malaysia's economic development, agriculture plays a vital role in providing employment opportunities in rural areas, raising rural income, and securing domestic food protection. The agricultural sector contributed RM455 billion (USD110.64 billion) to gross domestic product (GDP) with an annual growth of 2.4%, according to the Eleventh Malaysia Report (2016–2020). Since 2011, herbs have been listed as potential agricultural commodities within the National Key Economic Area (NKEA). In regard to the domestic Agriculture New Key Economic Area, the Malaysian government has selected the herbal industry as the first Entry Point Project (EPP1). The herbal industry's value in 2013 was around RM17 billion (USD4.13 billion) and is expected to rise between 8% and 15% annually to reach around RM32 billion (USD7.78 billion) by 2020 [34]. To support and ensure the fruitfulness of those programs, Herbal Cultivation Parks (HCP) was established in 2011 with the aim to produce a sufficient supply of natural herbal resources for clinical trials and research and development (R&D) prior to commercialize [35].

*Orthosiphon stamineus* Benth., from the family of Lamiaceae, is commonly referred to in Malaysia as Misai Kucing. The herb is popularly known in Southeast Asia as a remedy for eruptive, gallstone, hypertension, rheumatism, epilepsy, renal calculus, and syphilis [36]. Traditionally, the leaves of this plant are brewed and its herbal tea is consumed to enrich one's health while treating gout, diabetes, kidney, and bladder inflammation [37]. Furthermore, the plant has good antioxidant and anti-inflammatory properties that cause many researchers to examine the possible pharmacological characteristics of the plant, which are also anti-hypertensive, anti-tumoral, and anti-angiogenic [38–42]. Today, Misai Kucing is in great demand for herbal and pharmaceutical industry due to its medicinal [38] and economic value [15]. Othman et al. [6] conducted a study of inculcating Misai Kucing in solar PV farms. They have found that the size of Misai Kucing leaves under PV arrays is three times bigger than in normal conditions. Moreover, Misai Kucing under solar PV arrays grow vigorously, most probably due to the high soil moisture content compared with Misai Kucing cultivated under normal conditions. In other perspectives, Misai Kucing is classified as shade-loving plant. A previous study was carried out to investigate the effects of four different light levels (225, 500, 626, and 900 $\mu$mol m$^{-2}$ s$^{-1}$) imposed onto Misai Kucing. The results proved that Misai Kucing can survive under 225 $\mu$mol m$^{-2}$ s$^{-1}$ PPFD

and accumulation of secondary metabolites (such as Total Flavonoid and Total Phenolic) were more pronounced under low light levels (225 μmol m$^{-2}$ s$^{-1}$) which cultivated in the greenhouse [43].

However, up to now, there is no documentation or study report that has shown how these medicinal herbs grow in the shading conditions of the semi-transparent DSSC. Therefore, the purpose of this research was to determine the microclimate's temperature, relative humidity, Vapor Pressure Deficit (VPD), and Photosynthetically Photon Flux Density (PPFD) measured inside the semi-transparent DSSC shading greenhouse. Conjointly, this study targets to assess the effects of colored semi-transparent DSSC shading in altering the spectral of sunlight in reference to the Misai Kucing (*Orthosiphon stamineus*) growth responses. The photovoltaic greenhouse microclimate characteristics help determine the most suitable cultivation systems and assess strategies for increasing agricultural sustainability and electrical generation.

## 2. Materials and Methods

### 2.1. DSSC Greenhouse Pilot Model

Figure 1 illustrates the Portable Dye-Sensitized Solar Cell Mini Greenhouse that was developed using SOLIDWORKS software. The design's key compositions are the semi-transparent DSSC, adjustable racking system, collected water tray, castor wheel, sidewall with a slot, electrical and control system compartment, and fertigation system.

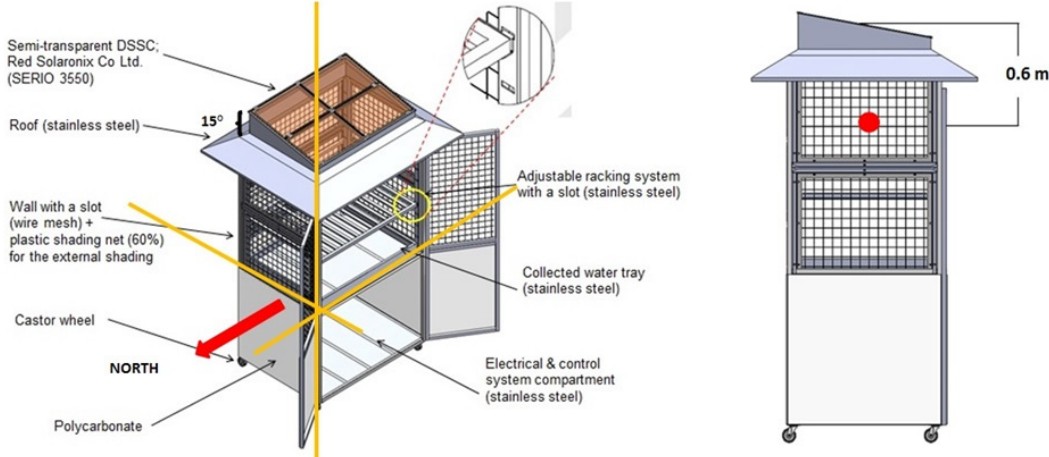

**Figure 1.** The detailed drawings of construction and compositions in Portable Dye-Sensitized Solar Cell Mini Greenhouse (PDMG) and the location temperature and relative humidity sensors inside the PDMG.

By installing the semi-transparent DSSC on the greenhouse's rooftop, it creates a layer of shading that affects the microclimate condition inside the greenhouse. To investigate the influence of this shading and the sunlight modification, a couple of identical mini DSSC greenhouse pilot models were put into test; the only difference for both is the installed semi-transparent DSSC panels as a rooftop where the model without it (glass as a rooftop) acted as the reference or control greenhouse. Both greenhouses had ample spacing between them in the same area to avoid intervention.

The dimensions of the greenhouses were 1.14 m long, 0.85 m wide and 2.18 m in height. Both greenhouses were in the same field in east–west orientation so that the plants would receive the same illumination resulting in successful cultivation. In addition, during the day, the shade from the mounted panels "moves" thereby preventing permanent shade of the plants. The glass and DSSC modules were fixed as a rooftop for both greenhouses (as shown in Figure 1). Both greenhouses' cover material was wire mesh with black net; 70%, (Henan Fengcheng Plastic, China). The black net available on the market cannot modify the spectral quality of sunlight and is completely opaque [32,44]. A greenhouse temperature of 35 °C or above is common, especially in lowland tropical climatic regions.

Such a warm condition can greatly affect crop development and vegetation [45]. Therefore, natural ventilation incorporated with the shading net is one of the solutions to reduce internal air temperature for a tropical greenhouse [46].

### 2.2. DSSC Panels

#### 2.2.1. DSSC Installation

In general, DSSC consists of dye (for light harvesting), substratum (Transparent Conductive Oxide—TCO), semi-conductor (Titanium Dioxide—$TiO_2$), counter electrode (Platinum/Carbon), and electrolyte (Triiodide—$I_3^-$/$I^-$, to invigorate the colorant and send the positive load to the counter electrode). There are four DSSC (red color) panels of SERIO 3550W19 (Solaronix, Aubonne, Switzerland). Each module has its dimensions of width, length, and thickness: 350 mm, 500 mm, and 3 mm, respectively. All modules were first fitted to the aluminum frame base. Then, all modules were connected in series and fixed on the east–west roof of the greenhouse. During the data collection, based on the UPM Software, Solar Noon Locator [47], the optimum angle of the DSSC panel was between 2° and 18° (April to May). Therefore, for these greenhouses, which were tilted at a 15° angle, they can get maximum solar radiation at noon and consequently generate the highest power, as reported by Khatib [48]. The area of the surface covered with DSSC (as the greenhouse rooftop) is 0.8 $m^2$.

#### 2.2.2. Electrical Characteristics of DSSC

The *I–V* electrical characteristics of the SERIO 3550W19 DSSC module are illustrated in Figure 2 (provided by Solaronix, Aubonne, Switzerland). The figure shows the red curve refers to the red DSSC module (SERIO3550W19) which was used in this study. The red DSSC module hits its highest peak electric power at 1.7 W with the solar radiation at 666 W/$m^2$. Previously studied by Roslan et al. [49] on electrical field performance of DSSC modules in the tropics, solar radiation increases, power produced is also steadily increases, and vice versa. The maximum power voltage value for 5 days was ranged between 1.783 and 2.081 W while the average power produced was 0.621 W.

**Figure 2.** The *I–V* electrical characteristics for Dye Sensitized Solar Cell (DSSC) modules, SERIO 3550W19 (red color). Source: from Solaronix, Switzerland.

### 2.3. Microclimatic Parameters Measurements

The investigation was carried out at the Faculty of Engineering, Universiti Putra Malaysia (UPM), Serdang, Selangor, Malaysia (3°0′30″ N latitude; 101°42′18″ E longitude; altitude of 63 m above sea level), with tropical-based weather conditions for 1 month; from

17 April 2019 to 17 May 2019. Malaysia experiences almost the same tropical weather conditions throughout the year. Thus, the duration of study on the influence of PV greenhouse on microclimate condition is considered sufficient since the climatic state of the country fluctuates throughout the year and some PV field studies backed it as appropriate [3,50,51].

The external and internal climatic conditions for both greenhouses, such as Photosynthetic Photon Flux Density (PPFD), relative humidity, and air temperature were measured for 1 month from 17 April 2019 to 17 May 2019. However, during this study, only sunny, hot, and clear sky periods have been presented, namely April 18, April 29, May 6, May 7, May 11, May 12, May 13, and May 17. All data were recorded for every 1-min interval by the logger. The middle point of the internal and external sides of both models was the points where the air temperature was measured using thermo-sensors connected to the logger (as shown in Figure 1). Three units of relative humidity sensors related to the logger were used to measure relative humidity, which were placed at the center of both greenhouses and outside. Meanwhile, the incoming PPFD ($\mu$mol m$^{-2}$ s$^{-1}$) was measured using three unit quantum sensors model 3668I, (Spectrum, Stanford, CT, USA), connected with WatchDog loggers (Spectrum, Stanford, CT, USA), respectively. One quantum sensor was located outside the greenhouse in a gutter height in order to prevent greenhouse shading. Another two quantum sensors were mounted at the center inside both greenhouses. All sensors' details are depicted in Table 1. The locations of the temperature and humidity sensor are illustrated in Figure 1.

**Table 1.** The details of the instruments.

| Instrument Model | Measurement Range | Accuracy |
|---|---|---|
| DS18B20 Thermo sensors, (Maxim Integrated, General Trias, Philippines) | −40–120 | ±0.5 |
| DHT-22 Temperature-Humidity sensors, (Aosong Electronics, Guangzhou, China) | 0–100 | ±2–5% |
| Catlog Series DAQ logger, (ARMX, Malaysia) | - | - |
| Model 3668I PAR Sensor, (Spectrum, Stanford, CT, USA) | 0–2500 $\mu$mol m$^{-2}$ s$^{-1}$ | ±5% |
| Model 400 Watch Dog logger, (Spectrum, Stanford, CT, USA) | - | - |
| USB2000 Spectrometer, (Ocean Optics, FL, USA) | 200–850 nm | - |

Vapor Pressure Deficit (VPD) is the correlation between the volume of water vapor and the retention capacity of water vapor saturated at the same given temperature and air [52]. VPD was calculated using the formula proposed by Jensen and Allen [53] as follows:

The saturation vapor pressure (SVP), $e_s$ resembles how much water vapor can be held.

$$\text{Saturation Vapor Pressure (SVP)}, \ e_s = 0.611 e^{\left(\frac{17.27 \times T}{237.3 + T}\right)} \tag{1}$$

with the units of vapor pressure (kPa) and air temperature, T (°C).

Actual Vapor Pressure (AVP), $e_a$ is included to resemble the humidity of the air in the actual weather.

$$\text{Actual Vapor Pressure (AVP)}, \ e_a = SVP \times \left(\frac{Relative \ Humidity}{100}\right) \tag{2}$$

The Vapor Pressure Deficit (VPD) resembles the correlation between how much water vapor can be held (SVP) and the humidity of the air (AVP) in the actual weather.

$$\text{Vapor Pressure Deficit (VPD)} = SVP - AVP \tag{3}$$

where Vapor Pressure Deficit (VPD) is in kPa.

The solar radiation spectra of the external and below the red semi-transparent DSSC were measured by USB 2000 spectrometer, (Ocean Optics, USA) and calibrated at noontime

of clear sky on 19 March 2019; at same location like the microclimatic measurement ($3°0'30''$ N latitude, $101°42'18''$ E longitude, altitude of 63 m above sea level). The fiber optic cable head connected to the spectrometer was pointed parallel to the sunbeam (for outside measurement) and 10 cm below the DSSC. All the data measurements appeared in Ocean View Software (spectrometer's USB cable connected with Laptop USB port).

### 2.4. Overall Thermal Transfer Value (OTTV) Calculation

Lu et al. [54] presented a novelty OTTV and heat transfer calculation on the integrated DSSC mini greenhouse (same prototype). The authors reported that the calculation of OTTV on DSSC mini greenhouse is comprised of two parts: walls (HDPE black shading net and stainless-steel wire mesh) and DSSC roof. With Malaysia Standard MS1525: 2014's OTTV equations as a guide, OTTV value for the walls contributes 92% (4442.68 $Wm^{-2}$) of the total OTTV of DSSC mini greenhouse while 16% (382.12 $Wm^{-2}$) OTTV is from the DSSC roof of the mini greenhouse. The OTTV calculation is computed in Table 2 as below:

**Table 2.** The novelty of the Overall Thermal Transfer Value (OTTV) calculation for the DSSC mini greenhouse studied by Lu et al. [54].

| Component of Greenhouse | Specific Material | Formula | $Wm^{-2}$ |
|---|---|---|---|
| Wall | HDPE black shading net; ($OTTV_1$) | $[15 \times \alpha_1 \times (1 - WWR)_1 \times U_1] \times A_1$ | 4441.5 |
| | Stainless steel; ($OTTV_2$) | $[15 \times \alpha_2 \times (1 - WWR)_2 \times U_2] \times A_2$ | 1.18 |
| | TOTAL OTTV (Wall) | $OTTV_1 + OTTV_2$ | 4442.68 |
| Roof | DSSC | $A_S \times U_S \times \Delta T + A_S \times SC \times SF\ A_S$ | 382.12 |
| | TOTAL OTTV (wall + DSSC roof) | $OTTV_{wall} + OTTV_{roof}$ | 4824.8 |

In this study, OTTV value for walls (HDPE black shading net and stainless steel) of DSSC mini greenhouse is exactly the same as used previously by Lu et al. [52] with a value of 4442.68 $Wm^{-2}$, since they are of the same prototype except for OTTV roof.

In this study, the roofs of both greenhouses are DSSC rooftop and glass rooftop, respectively. The OTTV formula for the rooftop of the greenhouse is as follows:

$$OTTV\ roof\ = \frac{(As\ \times\ Us\ \times\ \Delta T) + (As + SC + SF)}{As} \qquad (4)$$

Since this study has the same prototype (DSSC mini greenhouse) as previous studied by Lu et al. [54], some of the factors remain as follows:

- *As:* skylight area ($m^2$) = 0.7;
- *Sc:* shading coefficient. = 1 (No external shading device used for the DSSC and glass greenhouse);
- *SF:* solar factor for fenestration = 323;
- *Us:* thermal transmittance of skylight roof ($Wm^{-2}\ K^{-1}$). *Us* is defined by

$$U\ = \frac{1}{Rtotal}$$

where *R* is thermal resistance defined by

$$R\ = \frac{material\ thickness,\ l\ (m)}{thermal\ conductivity,\ k\ (Wm^{-1}K^{-1})}$$

Material thickness, (*l*), for the glass and DSSC rooftops (include frame base) are the same at 3.9 mm (0.039 m).

Thermal conductivity, (*k*), for DSSC and glass are 0.19 $Wm^{-1}\ K^{-1}$ and 1.05 $Wm^{-1}\ K^{-1}$, respectively. Thus, *U* values for the DSSC and glass rooftops are 48.72 $Wm^{-1}\ K^{-1}$ and 269.23 $Wm^{-1}\ K^{-1}$, respectively.

$\Delta T$ is defined as the difference value between outside and inside the greenhouse. All data were recorded by the temperature sensors for every 1-min interval by the logger for 8 days (same as the microclimate parameter data recorded). The average temperature difference $\Delta T$ for 8 days for the DSSC and control greenhouse are computed in Table A2.

*2.5. Plant Material and Growth Parameters*

Misai Kucing (*Orthosiphon stamineus*) was used as the test crop based on the previous study reported by Othman et al. [6] on solar PV farm. Misai Kucing was propagated by using stem cuttings in sand trays. When the Misai Kucing seedlings reached the stage of 6 to 7 true leaves (four weeks after planting), they were transferred into polybags filled with the composition of burnt rice husks, cocopeat, and chicken manure (proportion of 5:5:1) without any soil and acclimatized for 2 weeks. Each plant was fertigated with a nutrient solution that consisted of A & B Fertilizer (Formulation of Copper) as shown in Table A1. In terms of EC readings, the nutrient solution recorded readings from 1.5 to 2.0 μS/cm as measured by EC meters (AP-2, HM Digital, Seoul, Korea). The fertigation frequencies were set up for 2–3 min (150–250 mL) for 5 times per day by using timer (MST7, Kozuka, Honeywell, Tokyo, Japan). In order to determine the shading effect and solar radiation manipulation under the red DSSC, the plants were grown under two identical small-scale greenhouses; integrated semi-transparent DSSC greenhouse and control greenhouse (glass as a rooftop) started from 16 April 2019. The experiment was based on a t-test consisting of 12 plants in each greenhouse, totaling 24 subjects in the study. On week 12 of their cultivation, all these Misai Kucing plants were harvested. The growth variables, including the height of cultivations, number of leaves, amount of branch, and stem diameter were measured and collected. Calculation on the overall dry weight or the plant's total biomass was made by accounting for each seedling's leaves, stems, and roots of each seedling. Plant parts were extracted, inserted in a paper bag, and dried in an oven at 80 °C. An electronic weighing scale Model B303-S, (Mettler-Toledo Switzerland) was utilized to record the dry weight—a point where the plant parts' weight was unchanged. Meanwhile, the relative chlorophyll content of leaves (5 points/leaf) for 4 leaves for each plant was measured by a portable SPAD Meter Model 502 (Minolta, Tokyo, Japan). When the first white flower opened, the flowering period was noted.

## 3. Results

*3.1. Microclimatic Attributes*

3.1.1. Air Temperature

Table 3 exhibits the air temperature for both greenhouse and external records, specifically the lowest, highest, and mean readings. The maximum air temperatures for daily basis recorded inside the DSSC greenhouse, control greenhouse, and outside ranged between 30.5–38.0 °C, 40.5–43.5 °C, and 42.0–45.3 °C, respectively. Thus, the PV DSSC greenhouse marks a variation of as high as 5.5 °C in temperature with the control greenhouse, with a 7.3 °C maximum difference between internal DSSC greenhouses and outside. The mean temperature was the lowest for the DSSC greenhouse, followed by the control greenhouse and outside which were 31.1 °C, 32.5 °C, and 35.2 °C, respectively. Consequently, the daily average for air temperature inside the DSSC greenhouse was relatively lower (by approximately 1.2–2.0 °C) than the control greenhouse. These findings show that the DSSC acts as shading that prevents heat and minimizes the internal air temperature. Meanwhile, the minimum air temperature readings for both greenhouses internally were between 23.0 °C and 25.5 °C. The internal air temperature for both greenhouses match each other's readings, especially at night and early morning due to the natural ventilation inside these greenhouses, which help continuous fresh air movement inside the greenhouse.

**Table 3.** Lowest, highest, and mean air temperature readings for outside and within the DSSC and control (glass) greenhouses.

| Experiment Day | Control Greenhouse Temperature (°C) | | | DSSC Greenhouse Temperature (°C) | | | Outside Temperature (°C) | | |
|---|---|---|---|---|---|---|---|---|---|
| | Min | Max | Mean | Min | Max | Mean | Min | Max | Mean |
| 18 April 2019 | 24.5 | 43.5 | 32.6 | 24.5 | 38.0 | 30.7 | 26.4 | 44.2 | 34.9 |
| 29 April 2019 | 24.5 | 40.5 | 31.8 | 25.0 | 30.5 | 30.5 | 28.3 | 45.3 | 35.5 |
| 6 May 2019 | 24.0 | 42.0 | 31.1 | 24.0 | 36.5 | 29.9 | 26.2 | 44.7 | 35.3 |
| 7 May 2019 | 23.0 | 42.0 | 32.9 | 23.0 | 37.0 | 31.4 | 25.0 | 42.9 | 33.9 |
| 11 May 2019 | 23.5 | 42.0 | 33.8 | 23.5 | 36.0 | 32.1 | 26.7 | 45.0 | 36.6 |
| 12 May 2019 | 25.5 | 40.5 | 32.8 | 25.5 | 36.0 | 31.5 | 26.0 | 43.8 | 35.4 |
| 13 May 2019 | 24.5 | 41.0 | 31.7 | 24.5 | 36.0 | 30.5 | 25.3 | 44.4 | 34.3 |
| 17 May 2019 | 23.5 | 42.0 | 33.6 | 24.0 | 36.0 | 32.0 | 26.7 | 42.0 | 35.6 |

Figure 3 illustrates the difference between the external and internal air temperatures of the glass greenhouse (as control) and DSSC greenhouse. The record demonstrates upsurges in differences during daylight and reduction at night time. The air temperature differences (for daily basis) between the shaded DSSC greenhouse and un-shaded greenhouse (control) were high at noon, starting from 11 a.m. to 3 p.m. in the range of 4.5–10.0 °C. On the other hand, DSSC shading substantially reduced maximum temperature differences (daily basis) by 14.2% and 44.1% compared with the control greenhouse and outside, respectively.

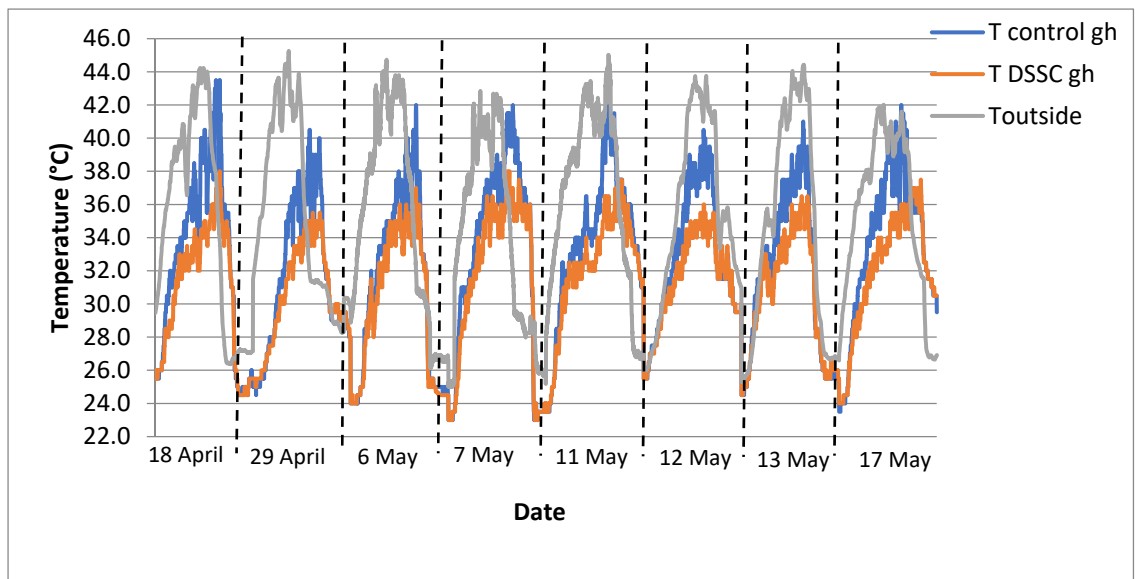

**Figure 3.** Air temperatures outside and inside greenhouses.

### 3.1.2. Relative Humidity

The presented graph shown in Figure 4 is the relative humidity (RH) for outside and inside both greenhouses (control and DSSC). RH is the air's water content measured up to the total quantity of water that can be held by air. According to this figure, the RH for outside and inside both greenhouse—DSSC and control—decreased during daytime, particularly afternoon on sunny days, then, increased during nighttime near to the maximum RH (100%). The decrease in the RH is a sign of increased air temperature during daytime and vice versa [55]. Moreover, the RH recorded inside the DSSC greenhouse was the highest, followed by the control greenhouse and outside, ranging between 97 and 52%, 98 and 44%, and 91 and 29%, respectively.

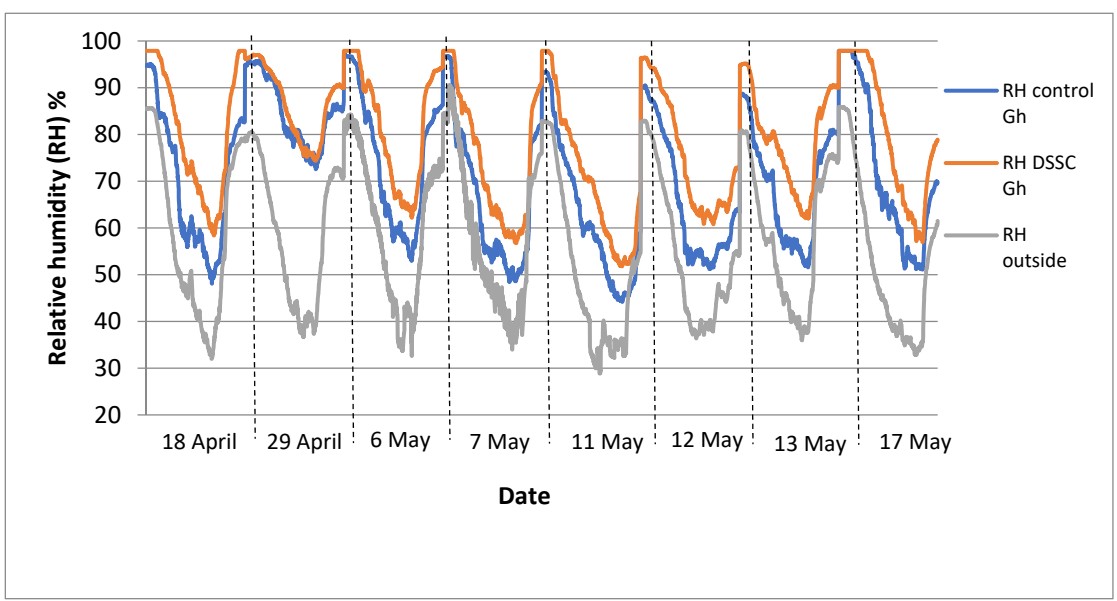

**Figure 4.** Relative humidity outside and inside greenhouses.

It was observed that the RH measured indoors (for both greenhouses) was consistently higher than outside during the middle of the day. Meanwhile, the average RH inside the DSSC greenhouse was 111% higher than the control greenhouse (glass) and 34% higher than the outside. The mean RHs for the outside, glass greenhouse (control), and DSSC greenhouse were 58.83%, 71.30%, and 79.08%, respectively. Additionally, during midday, the shading on the DSSC greenhouse has substantially increased RH differences (daily basis) by 13% and 45% compared with the control greenhouse and outside, respectively.

3.1.3. Vapor Pressure Deficit (VPD)

The plotted value of Vapor Pressure Deficit (VPD), as shown in Figure 5, justifies the "atmosphere" of plants with their environment in relation to the temperature and relative humidity. Thus, VPD measures the variation in air's water vapor level against its capacity when saturated at the same given temperature [52]. It was observed that the diurnal variation was very high during the day (10 a.m. to 5 p.m.) and decreased starting from late evening to early morning (6 p.m. to 8 a.m.). The maximum VPD (particularly at the central hour of the day) recorded outside varied between 4.76 and 6.03 kPa, while the maximum VPDs inside the un-shaded (control) and the shaded DSSC greenhouse were 1.911–4.42 kPa and 1.37–2.61 kPa, respectively. Meanwhile, the minimum VPD outside ranged between 0.34 and 0.95 kPa and the minimum VPD for internal control and DSSC greenhouse varied between 0.06 and 0.35 kPa and 0.06 and 0.15 kPa, respectively.

It was also observed that the indoor's VPD for both greenhouses were always lower than outside's VPD. The greenhouse material's thermal attributes resulted in the lower VPD inside both greenhouses compared with outside, where it reduced air temperatures inside and consequently increased the RH. Indeed, the average VPD for DSSC greenhouse was the lowest (1.07 kPa) followed by the control greenhouse (1.63 kPa) and outside (2.71 kPa). The low VPD inside the DSSC greenhouse may have been attributable to the shading effect of DSSC material. The shading effect does not only apply to the modification of the environmental conditions in greenhouses (e.g., lower the air temperature and increased the RH) but also can help to reduce VPD [56]. VPD reduction, in particular during the phase of solar radiation flux, significantly increased stomatal conductance and diffusion of CO2 concentration within plant; thus, it enhanced the photosynthesis rate of plant and agricultural productivity in greenhouse production [57]. In fact, VPD within the range of 0.8 kPa to 1.2 kPa is ideal for a greenhouse [58]. The VPD (1.07 kPa) inside the DSSC greenhouse in the present study is within the ideal range for a greenhouse. On top of that, alleviating the adverse effects, especially during midday atmospheric by DSSC shading,

has substantially reduced VPD by 34.2% (compared with control greenhouse) and 60.5% (compared with outside).

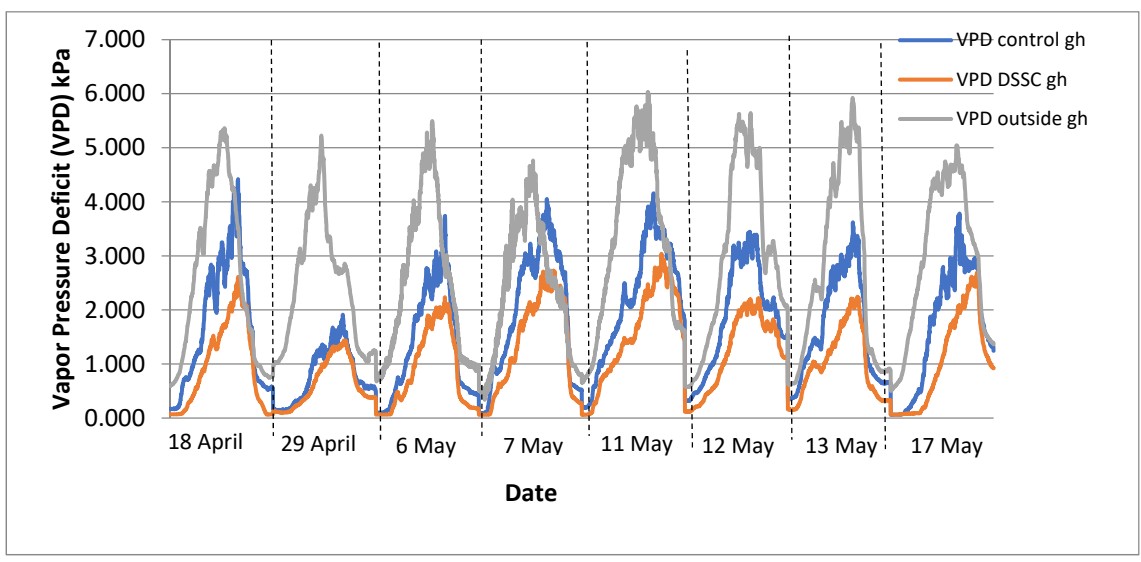

**Figure 5.** Vapor Pressure Deficit (VPD) outside and inside greenhouses.

### 3.1.4. Photosynthetic Photon Flux Density (PPFD)

The presented graph, as shown in Figure 6, is PPFD for the outside and inside the two greenhouses (DSSC and control). Results portrayed that the shaded DSSC greenhouse scored the lowest PPFD value compared to the greenhouse without shading and outside. Accordingly, the plants under the un-shaded greenhouse (control) persistently scored greater values of PPFD than the DSSC greenhouse. Remarkably, PPFD values under the two greenhouses at the early hours of the days were minimal compared with the outside. The PPFD amplified due to the sunlight until they achieved peak scores in the middle of the day (11 a.m. to 2 p.m.) when the sun reached its highest point.

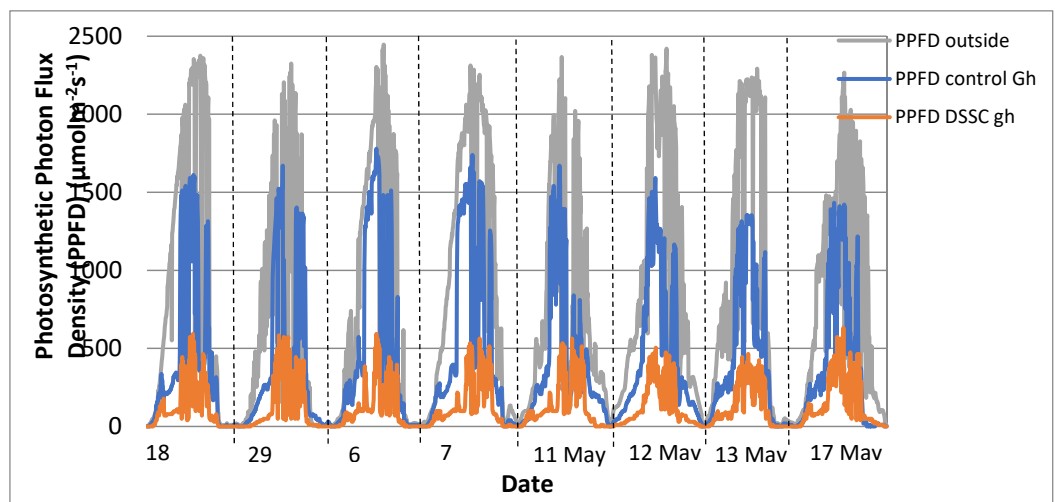

**Figure 6.** Photosynthetic Photon Flux Density (PPFD) values for outside, the control greenhouse, and the DSSC greenhouse.

The PPFD readings outside the greenhouse on bright and sunny days were strong, ranging from 2266.49 to 2445.86 µmol m$^{-2}$ s$^{-1}$ while internally the control greenhouse recorded a range of 1363.6 to 1798.4 µmol m$^{-2}$ s$^{-1}$ and lower for the DSSC greenhouse with a range of 336 to 474.3 µmol m$^{-2}$ s$^{-1}$.

### 3.2. Overall Thermal Transfer Value (OTTV)

Table 4 exhibits the OTTV results for wall and roof for both greenhouses. The total OTTV for DSSC and glass greenhouse is 4967.06 Wm$^{-2}$ and 5483.81 Wm$^{-2}$, respectively. The total OTTV for DSSC greenhouse is 4.94% less than control greenhouse. Major contributions of total OTTV, 89.44% and 81.01%, come from walls of DSSC and control greenhouse, respectively. Meanwhile, 10.56% and 18.99% come from the roof part of DSSC and glass greenhouses.

**Table 4.** The OTTV calculation for DSSC and control greenhouse.

| Component of Greenhouse | Specific Material | Formula | Wm$^{-2}$ |
|---|---|---|---|
| | HDPE black shading net; (OTTV$_1$) | $[15 \times \alpha_1 \times (1 - WWR)_1 \times U_1] \times A_1$ | 4441.5 |
| Wall | Stainless steel; (OTTV$_2$) | $[15 \times \alpha_2 \times (1 - WWR)_2 \, v \, U_2] \times A_2$ | 1.18 |
| | TOTAL OTTV (OTTV$_{Wall}$) | OTTV$_1$ + OTTV$_2$ | 4442.68 |
| Roof | DSSC (OTTV$_{DSSC \, roof}$) | $A_S \times U_s \times \Delta T + A_S \times SC \times SF \, A_S$ | 524.38 |
| | Glass (OTTV$_{GLASS \, roof}$) | | 1041.13 |
| Wall + Roof | DSSC greenhouse | OTTV$_{wall}$ + OTTV$_{DSSC \, roof}$ | 4967.06 |
| | Control greenhouse | OTTV$_{wall}$ + OTTV$_{GLASS \, roof}$ | 5483.81 |

### 3.3. Light Modification by DSSC Shading

The spectral irradiance distribution through DSSC shading and natural sunlight (outside) was measured while sunny at midday using spectrometer USB 2000 as illustrated in Figure 7.

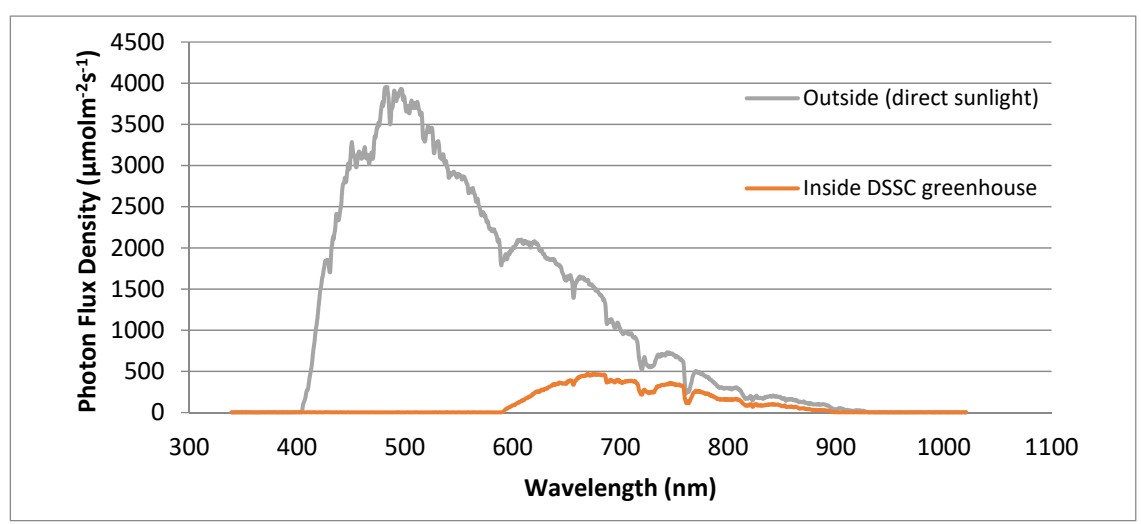

**Figure 7.** Spectral distribution outside and under integrated DSSC shading greenhouse.

The results revealed that red semi-transparent DSSC has a broad irradiance from the red region (600 nm) and above, and started to decline gradually in the far-red region from 826 nm. In addition, the red semi-transparent DSSC absorbs UV at the rate of 300–400 nm, blue region (400–500 nm), and green-yellow region (500–600 nm). In addition, the spectral irradiance of natural sunlight (outside) has a broad irradiance from UV up to the far-red region. However, the spectrometer used in this study is less sensitive to the UV region and the value was very small. The uniqueness of the semi-transparent DSSC is spectral manipulation (due to its variation of color as determined by the dye) and the scattering of transmitted light. Its ability to use diffuse light efficiently makes DSSC perform optimally despite low light settings, resulting in DSSC as a fantastic choice for greenhouse and windows (indoor purposes) [24]. Inada [59] states that the photosynthetic spectrum of energy hit the highest point in the red color range while only little for the blue color range

of various crops. Moreover, Oren-Shamir et al. [60] and Quail et al. [61] found out that phytochrome in photoreceptor systems functions to detect the red, far-red, blue, and UV light. Plants rely on this component to accurately detect and react to new light emergence.

There are extensive studies and approaches regarding spectral manipulations of sunlight on the red, far-red, and blue colors for greenhouses. Manipulations of light were achieved by using colored shade netting [60,62], colored fluid roof system [63,64], photoselective films (incorporated with pigments or dies) [65,66], and colored soil mulches [67,68]. For instance, Oren-Shamir et al. [60] tested varied shade nets in colors of black, blue, red, green, grey, and reflective, on cultivation growth and efficiency of Pittosporum variegatum. It has been found that the red colored net performed significant transmission of light of over 590 nm and the level of irradiance was slightly lower than natural sunlight. The red net has been shown to enhance branch elongation, photosynthesis rate, and leaf area of Pittosporum variegatum. From our study, it was also observed that the irradiance for DSSC was always lower than natural sunlight (outdoor), which may be attributable by the shading effect of DSSC. Previously, it was learned that the efficacy of radiation improves when the diffuse element of incident radiation is reinforced by shading [69]. Diffuse light is proven to improve crop productivity and boost ecosystem quality, while it additionally functions as a determinant in the period and quantity of flowering [70].

### 3.4. Plant Growth Results

*Orthosiphon stamineus* (Misai Kucing) grown under red DSSC shading tends to have a slightly higher number of branches, number of leaves, and total dry weight; but, it is slightly lower in plant height and stem diameter compared with Misai Kucing grown under un-shaded greenhouse as shown in Table 5 and Figure 8. Under DSSC shading, only light from red and far-red regions was transmitted while light from UV, blue, and green-yellow was absorbed (see Figure 7).

**Table 5.** Effect of light manipulation on growth of *Orthosiphon stamineus* (Misai Kucing) cultivated under glass (control) and DSSC greenhouses.

| | Plant Height | Number of Branch | Number of Leaves | Stem Diameter | Relative Chlorophyll Content | Total Dry Weight |
|---|---|---|---|---|---|---|
| | (cm) | | | (mm) | | (g/plant) |
| Control greenhouse | 38.69a $\pm$ 3.47 | 55.20a $\pm$ 7.66 | 260.44a $\pm$ 39.35 | 4.33a $\pm$ 0.42 | 31.70a $\pm$ 0.90 | 5.72a $\pm$ 0.95 |
| DSSC Greenhouse | 34.10a $\pm$ 1.58 | 58.40a $\pm$ 4.95 | 306.78a $\pm$ 38.77 | 3.55a $\pm$ 0.26 | 27.36b $\pm$ 1.40 | 8.24a $\pm$ 0.78 |

Data values with unique letters next to the values denote major variation at $p = 0.05$, $n = 18$.

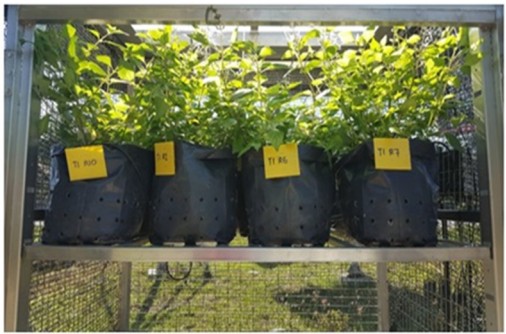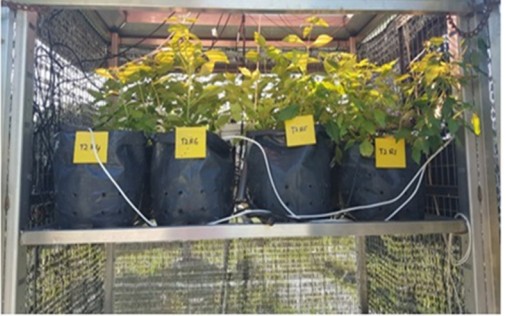

**Figure 8.** The test crop, *Orthosiphon stamineus* (Misai Kucing), under control greenhouse with glass as a rooftop (**left picture**) and integrated semi-transparent DSSC as a rooftop (**right picture**).

It was also observed that the relative chlorophyll content (SPAD unit) of Misai Kucing under DSSC shading greenhouse was significantly lower compared with the un-shaded greenhouse (as shown in Table 5). The low relative chlorophyll content under DSSC greenhouse is probably due to the shading effect of DSSC. A previous study has proved that under moderate shade level, the chlorophyll content had significantly decreased then increased again at heavy shade levels [71]. In general, chlorophyll content increases with increasing shade level. However, in this study, DSSC shading can be classified as moderate shade level which reduced chlorophyll content in plants.

In this study, delay flower initiation of Misai Kucing is observed under red DSSC shading greenhouse, as shown in Figure A1. Mean days to flower for Misai Kucing grown under un-shaded and red DSSC greenhouses were 31.2 and 34.5, respectively. The first harvest of Misai Kucing for normal practices is 10 weeks after planting in the fields and just before flowering (due to the high phytochemicals content). Therefore, the delayed flower initiation of Misai Kucing cultivated under red DSSC shading greenhouse could be beneficial for the quality of this medicinal herb due to its high phytochemical content and high antioxidant activities.

## 4. Discussion

The low temperature inside the DSSC greenhouse has similar results found in a study by Hassanien et al. [18], whereby semi-transparent photovoltaic (mono-crystalline silicon) acts as shading reducing the air temperature approximately by 1–3 °C under natural ventilation. This shading effect is an ideal condition for plants during the excessive temperature periods, especially afternoon. The effect does not apply only to the microclimate modification (such as reduction of air temperature), but also can help to minimize thermal efficiency, operating system phases, transpiration, water, as well as electric consumption [56,72]. The thermal attributes of the rooftop composition, shading effects, and plant development may explain the difference in air temperature between the DSSC greenhouse and control greenhouse (glass) [55]. The results also indicate that the internal air temperature of DSSC greenhouse was consistently cooler compared with the external temperature and the control greenhouse without shading. This applied shading managed to reduce the air temperature by almost 5.9 °C compared to the un-shaded greenhouse on hot and sunny days. Hassanien and Li [1] reported similar findings. Their research involved a partially translucent mono-crystalline silicon PV fixed on the rooftop of a greenhouse that took place in microclimate conditions where they focus on the effects of this attachment. The air temperature under Building Integrated Photovoltaic (BIPV) greenhouse decreased approximately 1.0–3.0 °C compared with the un-shaded greenhouse. Nevertheless, relative humidity is not influenced by natural ventilation.

The high RH inside the DSSC greenhouse is due to the shading effect of DSSC. Shading is commonly applied to cut down the intense solar radiation entering the greenhouse. The shading reduces the rate at which irradiation is converted to heat in the greenhouse [72,73]. In such a condition, the greenhouse air temperature will be decreased and consequently increase the relative humidity. Plant cultivation would be right and proper within a 60–90% range of RH [74]. Lower RH of below 60%, particularly during the daytime on a sunny day, can cause water stress resulting in closed plant stomata and constrained respiration. As a result, the photosynthesis, as well as physiological process of plant, may be disrupted and significantly impact crop growth. In another situation, if the RH exceeds 95%, particularly at night, it interferes with plant transpiration and fosters the development of fungal diseases and insects breeding. Moreover, lower transpiration (due to the excessive RH) can cause condensation of droplets on the walls and ceiling, which reflect sunlight.

VPD is an ideal indicator of plant stress whereby it measures potential water stress within a plant. An escalating VPD signifies the decreasing RH while the decrease in the VPD is a sign of increased RH. In this study, the shading of DSSC reduced the indoor VPD of greenhouses. Mashonjowa et al. [75] investigated the effect of shading on VPD and temperature of a multi-span greenhouse. This greenhouse, specially equipped with Low

Density Polyethylene (LDPE) film (250 ηm) glazing, exhibited maximum air temperature variation of the greenhouse's exterior and interior, before and after shading were 2.4 °C and 1.6 °C, respectively. In addition, the VPD was reduced 2 kPa after shading the greenhouse. Generally, excessive VPD (approximately 2 kPa) contributes to excessive transpiration and risking from being well-watered plants. In a worse situation, plants may wither and photosynthesis can be extremely reduced [55,76]. In contrast, the transpiration process may be suppressed if VPD is too low (less than 0.2 kPa), causing plants to transpire inadequate water to carry mineral nutrients (e.g., Calcium) through the xylem. Moreover, when the VPD is extremely low (95–100% RH), water may condense onto the plant and encourage disease and fungal growth.

Most of the heat gain for walls contributes from high heat conduction through High Density Poly-Ethylene (HDPE); it means in this condition the *U*-value is high. Moreover, a significant effect on heat conduction through HDPE is caused by absorptivity of black HDPE [54]. In this study, the total OTTV for DSSC greenhouse was lower compared with control greenhouse. The shading effect of DSSC (contains dye) reduces transmissivity of solar radiation, thus reduces the solar heat gain inside the DSSC greenhouse. OTTV is a useful tool to measure the overall heat transferred into a structure's layer, especially for a greenhouse [77]. This is because, in tropical climate regions like Malaysia, the high irradiation, temperature, and vapor pressure deficit cause heat stress and detrimental effects on tropical crop production. In addition, those conditions are more pronounced in an enclosed greenhouse structure. Thus, information on the OTTV value, especially on the greenhouse, is vital to grasp its thermal system, leading towards the implementation of an effective cooling system [78].

The semi-transparent DSSC can be positioned in between two spectrums of technology: the solid, non-transparent PV (mono-crystalline silicon) and glass cover as semi-transparent DSSC has higher light transmission than the opaque PV and performs weaker than the glass cover of greenhouse. Moreover, semi-transparent DSSC can easily be fitted in for certain greenhouses, especially those that cultivate herbs. This study observed that DSSC shading has substantially reduced PPFD by 75.8% (compared with the control greenhouse) and 92.2% (compared with outside). The locally known Misai Kucing or *Orthosiphon stamineus* is a type of shade-loving plant. It does not require a high light intensity of sunlight. The previous study by Othman et al. [6] proved that cultivating Misai Kucing under solar PV panel (mono-crystalline) gives strong financial return and Misai Kucing grew more vigorously than in normal conditions (without PV). In addition, previous research proved that Misai Kucing can survive at 225 $\mu$mol m$^{-2}$ s$^{-1}$ PPFD and accumulation of secondary metabolites was also found to be more pronounced under low light levels [43].

Plants are capable of recognizing gradual shifts in the direction, structure, and intensity of light within their maturation ecosystem. For that reason, plants could potentially respond to these shifts physiologically, biochemically, and morphologically to withstand the current environmental state. Plants use phytochrome to capture these light signals and the light spectrum (UV, red, far-red, and blue regions) are vital elements in this cycle [28]. Since light from red and far-red regions was transmitted under red DSSC shading, only red light reaches the plants, while far-red light harnessed the electricity generation. Therefore, the ratio of red light with the far-red light (R:FR) radiation generated by red DSSC shading in the present study is higher compared with R:FR under un-shaded greenhouse (control) and natural sunlight. Previous research reported that red light has been shown to stimulate overall vegetative growth while far-red light, on the other hand, nullifies the mediated effects of red light [28,70]. To briefly summarize, red light demonstrates the capacity to enhance branches' number and development [60,79], increase the number of leaves [80], and enhance biomass production (fresh and dry weights) [81]. Far-red light has been shown effective in reducing stem elongation and plant height [28,70,82], resulting in being smaller in size [79,80] and delayed time of flowering [79].

## 5. Consideration and Future Development of DSSC Greenhouse

The advantage of integrated semi-transparent DSSC greenhouse is mainly related to the agronomic sustainability compared with conventional PV greenhouse (crystalline silicon-based). This study demonstrates that the growth of Misai Kucing under the DSSC greenhouse was acceptable and the same like Misai Kucing cultivated under normal (control) greenhouse. Even though the plant growth results of Misai Kucing cultivated under the DSSC mini greenhouse and the control greenhouse are almost the same, this study highlights the concept of agri-voltaic (AVS) (mixed system combining solar PV and crops simultaneously in the same land area) whereby they help reconcile food security and the supply of green energy. With this concept, farmers would simultaneously harness revenues driven from agricultural activities and the generation of electric power. Agricultural activities incorporating PV technologies not only promote the introduction of renewable energies, but also support energy conservation and environmental concerns. Compared with conventional PV greenhouse, the crystalline silicon PV module is opaque to sunlight, which is why its crop productions are poorly inflicted by the modules shading. In fact, several previous studies reported significantly negative findings of shading effects on Welsh onion [83], tomato [11], lettuces [84], peach, and cherry [85]. The integration of the semi-transparent PV panel as the rooftop of this conventional PV greenhouse serves as an apparent alternative to the substandard crops production and quality.

The unique properties of DSSC, such as light transmissivity and various colors determined by the dye, encouraged the optimal physiological reactions from plants (higher in the number of branches, leaves, and plant biomass), through the adjustment on the spectrum of light and the reduction in the usage of chemical, cost of labor, and plant hormones. DSSC is also comparably more attractive and holds aesthetic value than the conventional PV, especially for indoor-purposed structure [21]. Additionally, the weight and rigidity of crystalline silicon modules are easily outclassed by the lighter and elastic DSSC, making them useable with a large number of plastic-made greenhouses worldwide [21]. DSSC technology seems to offer the answer to the suitable material for the existing PV greenhouses improvisation; on the issue to withstand the weighty crystalline silicon modules. Although DSSC and other semi-transparent solar cell technologies have been exclusively designed to be integrated with the greenhouse, every so often they are costly and need extensive research and development to deliver the effectiveness and durability in the farm and outdoor climate. Moreover, problems such as leaked electrolyte, thermally unstable dye molecules, and electrocatalytic activity of the counter electrode (CE) interface, subsequently lowers the lifetime of DSSC and taking a toll on its performance [86–88].

Thus, researchers must resolve the DSSC stability and efficiencies for Building Integrated Photovoltaic (BIPV) applications (i.e., greenhouse, rooftop building, etc.) by developing perfect encapsulation, more stability dyes, less volatile electrolytes with stabilizing additives, and quasi or completely solid-state carrier mediators [89]. Typical DSSC electrolyte leakage, triiodide $I^-/I_3^-$ (liquid electrolytes), may be replaced by solid and polymer electrolytes determined to seal and eliminate the leakage of solvent problems. On the other hand, the changes in platinum's electrocatalytic properties as a CE (i.e., the valence state) may not be constant over a long period. Therefore, carbon materials are the new latent alternative for platinum materials that can replace platinum due to electronic conductivity, thermal stability, electrochemical stability, high specific area, and high mechanical tolerance [90]. Additionally, UV light has also adversely affected electrolyte stability. To prevent side reactions, $MgI_2$ is added as a solution to re-surface the external of MgO. With this addition, it resulted in a 3300 h of stability at 2.5 sun [91]. However, further research regarding DSSC efficiency and stability is still required.

## 6. Conclusions

The integration of semi-transparent DSSC shading greenhouse is an innovative way of reconciling energy generation by photovoltaic systems and agricultural activities. Since traditional greenhouse PV (silicon material) is impervious to sunlight, a greenhouse integrated with semi-transparent DSSC shading appears valuable to minimize the impact on crop growth, especially in a tropical climatic condition like in Malaysia. Moreover, the uniqueness of DSSC shading by spectral manipulation (due to its color can be determined by the dye) is a palliative way against chemical growth regulator uses which has severely affected humans and the environment. The shading effect of semi-transparent DSSC affects the greenhouse's microclimate and the Misai Kucing production. From this study, semi-transparent DSSC shading lowered interior heat by 1.47 °C and increased the relative humidity by 10.91% on central hours of the day compared with the un-shaded greenhouse. Moreover, the average VPD for DSSC greenhouse was 1.07 kPa, which is an ideal VPD in a greenhouse compared with the un-shaded greenhouse of 1.63 kPa, which may influence plant stress. Since Misai Kucing has a characteristic of a shade tolerant plant, the maximum PPFD ranging from 2266.49 to 2445.86 $\mu mol\ m^{-2}\ s^{-1}$ for outside and 1363.6 to 1798.4 $\mu mol\ m^{-2}\ s^{-1}$ for un-shaded greenhouse may disrupt Misai Kucing, especially at the central hours of a sunny day. Thus, the application of DSSC modules as shading is an ingenious way to create at least near the optimum microenvironment needs of plants. In addition, the growth of Misai Kucing cultivated under the DSSC greenhouse was acceptable and is the same as Misai Kucing cultivated under normal (control) greenhouse. This study highlights the concept of agri-voltaic (AVS) (combination of PV system and agriculture within same space or land unit area) whereby this system helps to reconcile food security and at the same time supply the green energy. In addition, the delayed flower initiation of Misai Kucing cultivated under DSSC shading greenhouse may be beneficial for the quality of this medicinal herb due to its high phytochemical content and high antioxidant activities. Further research is required to investigate the effects of spectrum manipulation on physiological and bioactive compounds of these medicinal herbs. Moreover, the study on the performance of various DSSC transparencies is also necessary in pursuit of high energy generation.

**Author Contributions:** Conceptualization, N.R. and M.E.Y.; methodology, A.N.I.; software, M.H.O.; validation, M.E.Y., D.J., and Y.H.; formal analysis, N.R.; investigation, N.R. and J.M.; resources, N.R.; data curation, M.R.A., M.H.I., M.F.M., and B.S.N.A.; writing—original draft preparation, N.R.; writing—review and editing, N.R., M.E.Y., D.J., and Y.H.; supervision, A.H.J.; resources and statistical analysis, M.E.Y. and D.J.; funding acquisition, M.E.Y. and D.J.; OTTV calculation in methodology, L.L. All authors have read and agreed to the published version of the manuscript.

**Funding:** This research was funded by Research Management Centre (RMC), Universiti Putra Malaysia for the approval of research funding under the IPB Putra Grants Scheme (Vote no: 9515303) and the Malaysia Electricity Supply Industry Trust Account (MESITA) (Vote No. 6300921).

**Institutional Review Board Statement:** Not applicable.

**Informed Consent Statement:** Not applicable.

**Data Availability Statement:** Data will be made available upon publication.

**Acknowledgments:** The authors delegate our thanks to the Centre of Advanced Power and Energy Research (CAPER), Research Management Centre (RMC), Universiti Putra Malaysia.

**Conflicts of Interest:** The authors declare no conflict of interest.

## Appendix A

**Table A1.** Nutrient solution formulation and chemical characteristics applied in the study [92].

| Fertilizer/Salt | Formula | Weight of Salt g/20 L |
|---|---|---|
| **Stock A** | | |
| Calcium nitrate | $Ca(NO_3)2.4H_2O$ | 10,003 |
| EDTA Iron | $CH_2N (CH_2.COO_2)2Fe Na$ | 790 |
| **Stock B** | | |
| Potassium dihydrogen orthophosphate | $KH_2PO_4$ | 2630 |
| Potassium nitrate | $KNO_3$ | 5830 |
| Magnesium sulphate | $MgSO_4.7H_2O$ | 513 |
| Manganous sulphate | $MnSO_4.H_2O$ | 61 |
| Boric acid | $H_3BO_3$ | 17 |
| Copper sulphate | $CuSO_4.5HO$ | 3.9 |
| Zinc sulphate | $ZnSO_4.7H_2O$ | 4.4 |
| Ammonium molidate | $(NH_4)_6MO_7O.4H_2O$ | 3.7 |

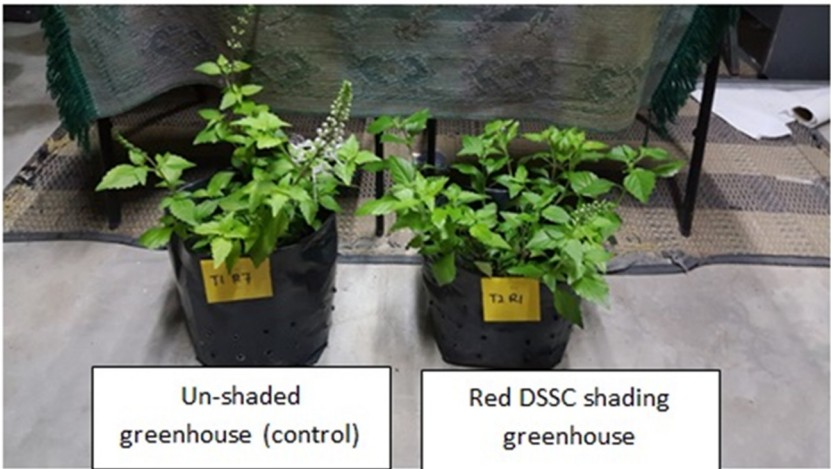

**Figure A1.** The morphological reactions of Misai Kucing cultivated under un-shaded (control) and red DSSC shaded greenhouses 10 weeks after planting.

**Table A2.** $\Delta T$ (the difference value between outside and inside) for both greenhouses: DSSC and control greenhouse.

| | 18 April | 29 April | 6 May | 7 May | 11 May | 12 May | 13 May | 17 May | Average |
|---|---|---|---|---|---|---|---|---|---|
| **$\Delta T$ DSSC Gh (°C)** | 4.22 | 5.00 | 5.40 | 2.50 | 4.54 | 3.93 | 3.87 | 3.59 | 4.13 |
| **$\Delta T$ control Gh (°C)** | 2.27 | 3.78 | 4.25 | 1.02 | 2.86 | 2.60 | 2.63 | 1.93 | 2.67 |

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
