# Peer review of "Dye-Sensitized Solar Cell (DSSC): Effects on Light Quality, Microclimate, and Growth of Orthosiphon stamineus in Tropical Climatic Condition"

_agronomy, doi:10.3390/agronomy11040631_

Round 1

Reviewer 1 Report

The authors adress an interestig topic. The report of the is addecuated but several issues must be corrected:

Main problems:

  • Please follow the recomended structure of Agronomy: In your manuscript Results and discussion are one section where two different sections would facilitate the lecture
  • Several information is repeated along the manuscript. Please review it to either explain it in introduction and avoid such an extensive manuscript
  • The title of the manuscript and goal, is to report the environmental effect of DSSC cells, therefore the last part of your manuscript about the econmical importance need to be reviewed, rewrited and reduced as is relatively long as is not part of the main goal.
  • In general all environmental variables figures could be plotted together , using subfigures.
  • Finally, the conclusions are pushed to a positive effect of your treatments. Sadly, this is a common problem and with this data you were not able to prove a significant effect of the usage of DSSC cells in the production of Orthosiphon spp. There were clear indications but the used sample size would need to be increased to prove it.

Minor comments:

51: CO2 . Please use subscripts.
123-138: As this is an international journal, I would recomend to use either USD or EUR.
140: Remove figure 1. The Paper is long enough and is not necessary.
141: Latin names in italic
156: 225 umol is actually not a low level of light. Please rephrase
189: I woudl recomend to remove Figure 3 . There is already a diference between treatments, where the control doesn't have a close compartment bellow the unit...
199: Missing information. (the light trans-199 mittance of the shading net, brand, model and country of manufacturing)
205: Figure 4 is a partial duplicate of figure 1. Please remove or complement
215: When refering to a manufactoring please indicate in parentesis (Company, country). It need to be consistent along the manuscript.
218: Repited information (15°). It was already discussed. Please remove
225: Figure 5 is unnecesary. Please remove.
230-232: Please rephrase. Interpretation of the figure is inadecuate.
242: Table 1 not necessary
243-247: Please rephrase. The average power cannot varied. is an average. Neither cannot have an specific time.
Figure 6 : Please modifify to have same format that other figures or remove, as doesn't bring important information.
Figure 7: This should be presented as results, not material and methods if was measured during this study. Any reason to choose these days? if not, please remove and indicate range of values.
251-268: Same here. This is partialy results
298: Figure 8: Please remove. as previusly, could be merged with figure 1.
317: If spectrometer was calibrated using sunlight, please indicate day of the year and geographic cordinates
340: Table 4: When refering to chemical components, please use sub indexes.
342: Figure 9: please move to results.
352: In how many leaves?
369- 374: Move to discussion
390-401: Move to discussion
423-438: Move to discussion
452-460: Move to discussion
496-501: Move to introduction
519-538: Move to discussion
583-604: Move partialy to introduction and discussion.

606: Please add s.e or s.d 
626: Figure 15: I would suggest to move to supplementary figures.
643-645: This is actualy statistically wrong. You were not able to prove it and if desired, a bigger sample size would be need it

From 683 onwards: (Table7,8,9, Figure 16) I suggest to removed it. THe goal of the paper was to characterize the weather udner DSSC technology, not an econmical study.

768-770: Wrong conclusion. There were indications but the used sample was too small. 
780-783: Remove

Author Response

Thank you for the effort and comments given. We have really considered the concerns and complete the revision as enclosed doc. The abstract and conclusion are also revised accordingly. Thanks

Reviewer 2 Report

Agronomy

Manuscript Number: agronomy-1134560

Title: Dye-Sensitized Solar Cell (DSSC): Effects on Light Quality,

Microclimate and Growth of Orthosiphon stamineus in Tropical Climatic

Condition

This manuscript reports tropical plant (Orthosiphon stamineus) cultivation results obtained for a mini greenhouse, the roof of which was replaced with dye-sensitized solar cells (DSSC). The use of greenhouse roofs for PV deployment is gaining wide attention recently. Although several reports of the scientific literature have described some benefits of greenhouse-integrated DSSCs, they have primarily emphasized material development aspects. Given the state of current research, the benefit of a study by Roslan et al. (agronomy-1134560) is that it elucidates causality between crop growth and microclimate in the test greenhouse in which DSSC modules were installed. The authors have explained plant response adequately under DSSC shading in terms of the greenhouse microclimate. Although some improvements are still necessary, the use of DSSCs as a shading material will penetrate into tropical greenhouses in which sunlight and temperature levels are often excessive. For these reasons, this reviewer believes that the data reported by the authors are invaluable and original. Nevertheless, some concerns remain in the manuscript, as explained below.

  • The time axes (horizontal axes) in Figs. 10-13 should be revised properly because presented time data are discontinuous (18 and 29 April, 6, 7, 11, 12, 13, and 17 May). The insertion of gaps between the discontinuous dates is recommended.
  • The unit of spectral irradiance in Fig. 14 is questionable. If the unit is truly in its current form, then an explanation must be given in the manuscript to explain why the unit of the spectral irradiance is given by the authors as micromoles per square meter per second.

Author Response

Thanks for the comments and effort for considering the manuscript in Agronomy. We have done the revision as per attached doc. thanks
